# Dysregulated Cell–Cell Communication Characterizes Pulmonary Fibrosis

**DOI:** 10.3390/cells11203319

**Published:** 2022-10-21

**Authors:** Jonathan S. Kurche, Ian T. Stancil, Jacob E. Michalski, Ivana V. Yang, David A. Schwartz

**Affiliations:** 1Department of Medicine, University of Colorado Anschutz Medical Campus, Aurora, CO 80045, USA; 2Rocky Mountain Regional VA Medical Center, Aurora, CO 80045, USA; 3Program in Cellular Biology and Biophysics, Graduate School, University of Colorado Anschutz Medical Campus, Aurora, CO 80045, USA; 4School of Medicine, University of Colorado Anschutz Medical Campus, Aurora, CO 80045, USA

**Keywords:** mucin, *MUC5B*, rs35705950, type-I alveolar epithelial cell, fibroblast, alveolar macrophage, idiopathic pulmonary fibrosis, cell–cell communication, IL6, AREG, ADAM17

## Abstract

Idiopathic pulmonary fibrosis (IPF) is a progressive disease of older adults characterized by fibrotic replacement of functional gas exchange units in the lung. The strongest risk factor for IPF is a genetic variantin the promoter region of the gel-forming mucin, *MUC5B*. To better understand how the *MUC5B* variant influences development of fibrosis, we used the NicheNet R package and leveraged publicly available single-cell RNA sequencing data to identify and evaluate how epithelia participating in gas exchange are influenced by ligands expressed in control, *MUC5B* variant, and fibrotic environments. We observed that loss of type-I alveolar epithelia (AECI) characterizes the single-cell RNA transcriptome in fibrotic lung and validated the pattern of AECI loss using single nuclear RNA sequencing. Examining AECI transcriptomes, we found enrichment of transcriptional signatures for IL6 and AREG, which we have previously shown to mediate aberrant epithelial fluidization in IPF and murine bleomycin models. Moreover, we found that the protease ADAM17, which is upstream of IL6 trans-signaling, was enriched in control *MUC5B* variant donors. We used immunofluorescence to validate a role for enhanced expression of ADAM17 among *MUC5B* variants, suggesting involvement in IPF pathogenesis and maintenance.

## 1. Introduction

Idiopathic pulmonary fibrosis (IPF) is an incurable disease of heterogeneous, progressive, lung parenchymal fibrosis affecting over 5 million people worldwide [1]. Prevalence of IPF is estimated to be anywhere from 0.02–0.04% of the population, and up to 2% over the age of 50 in higher risk populations [2]. While recent advances have yielded some therapies that appear to slow the rate of progression [3,4], median life expectancy of individuals with IPF remains 3–5 years [1]. Importantly, age is a critical determinant of IPF, such that the odds of biopsy-proven IPF among patients referred to a pulmonary clinic for suspected interstitial lung disease are increased by almost 10% per year of life (OR, 1.09 per year, 95% CI 1.04–1.14, *p* = 0.0007) [5]. Over the past decade, we discovered a gain-of-function [6] promoter variant in *MUC5B* (rs35705950, G; T) that is the dominant risk factor for IPF (OR = 5.45; 95% CI = 4.91–6.06; *p* = 9.60 × 10^−295^) [7], present in >50% of affected patients and accounting for at least 30% of the risk of disease [8,9]. These findings have been validated in more than 11 independent studies [7,8,9,10,11,12,13,14,15,16]. In IPF, MUC5B is expressed in areas of dense fibrosis [17,18,19]. Moreover, ectopic alveolar expression of Muc5b enhances the development of bleomycin-induced lung fibrosis in mice [20,21]. Augmented airway clearance with a novel mucolytic reduces durable fibrosis even long after bleomycin injury [20], supporting a role for secreted MUC5B in IPF.

In spite of these advances, the role of the *MUC5B* variant on promotion of fibrosis remains unclear. We recently demonstrated IPF airway epithelia are marked by an aberrant fluidized phenotype in culture associated with excess EGFR signaling mediated by AREG [22]. Moreover, we have recently confirmed and extended findings that Il6 signals are necessary for fibrotic lung remodeling in the setting of bleomycin injury in the mouse [23,24]. To better understand the role of cell signaling involved in *MUC5B*-variant-driven pulmonary fibrosis, we used publicly available single-cell and single nuclear RNA sequencing datasets (“Vanderbilt”, GEO accession GSE135893 [25] and “Colorado”, GSE161685 [26]). We observed that disappearance of type-I-alveolar epithelial cells (AECI) characterized fibrotic lung disease at the single-cell and single nuclear level. We reasoned that loss of AECI might be mediated by changes in cell signaling within the alveolar microenvironment. Focusing on cell–cell communication in AECI, we identified gene signatures of active ligands in datasets from IPF and control tissues using the R software platform NicheNet [27]. Our analysis revealed signatures of growth factors, cytokines, and matrikines influencing AECI transcriptomes in IPF, including IL6, IL1B, TNFA, AREG, FGF2, FGF7, SPP1, and TRAIL. Moreover, we observed enrichment for ADAM17-dependent signaling in AECI from *MUC5B* variant vs. nonvariant donors and observed differential ADAM17 protein expression in separate variant-derived tissues. Taken together, the analysis presented here supports a role for ADAM17 and its substrates in AECI loss in *MUC5B*-variant driven IPF.

## 2. Materials and Methods

### 2.1. Analyzed Datasets

The standard single-cell CellRanger (10X Genomics, Pleasanton, CA, USA) output files for the IPF single-cell datasets (GSE135893, derived from investigators at Vanderbilt University [25] and GSE161685, derived from our group at the University of Colorado [26]), were imported into the R programming environment (R4.1.3, The R Foundation for Statistical Computing, 2022) using the Seurat software package (version 4.1.1, https://satijalab.org/seurat/, New York, NY, USA, accessed on 19 June 2022 [28]). These files represent deidentified, demultiplexed, aligned, single-cell and single nuclear next-gen sequencing output from Illumina HiSeq 4000 or NovaSeq 6000 (Vanderbilt dataset), or Illumina HiSeq 4000 or NextSeq 500 (Colorado dataset). The Vanderbilt samples are composed of deceased donor lung specimens rejected for transplant or IPF lungs explanted at time of transplant at the Norton Lung Institute in Phoenix, AZ, or Vanderbilt University. For control lungs, multiple regions within 2 cm of the pleura were harvested for single-cell digestion. For IPF lungs, multiple regions representing areas of disease involvement were harvested [25]. For details on available patient demographics in the Vanderbilt dataset, see Appendix A. The Colorado dataset samples are composed of whole lung explants or peripheral biopsy specimens from the University of Pittsburgh or the Lung Transplant Research Consortium, flash frozen from buffered medium, and stored at −80 °C from the time of harvest. Participants or kin were consented for genetic studies at the time of harvest. Frozen tissue fragments were taken from the serosal tissue surface when identifiable on the frozen specimen [26]. For details on patient demographics in the Colorado dataset, see Appendix A.

### 2.2. Single-Cell and Nuclei Sequencing Analysis

Single-cell samples from the Vanderbilt dataset were filtered for RNA count per cell (1000 < RNA count < 30,000) and mitochondrial content (mitochondrial genes < 10% of total), and then selected for diagnosis of IPF or normal/control. Single nuclear samples from the University of Colorado dataset were filtered for a slightly lower RNA cutoff, reflecting the expected enrichment of RNA in nuclei (500 < RNA count < 25,000) and mitochondrial content (<10% total genes). Samples were normalized using the SCTransform v2 function (version 0.3.3, https://github.com/satijalab/sctransform/, New York, NY, USA, accessed on 18 June 2022) [29], which runs an L1 (lasso)-regularized, variance-stabilized, negative binomial regression against sequencing depth for any given barcode/cell in the dataset and uses the residuals to perform the normalization. Scaling, clustering, and downstream differential expression were performed by running the “PrepSCTFindMarkers” function in the dataset. We used the “v2” flag for “vst.flavor” and selected the top 30 principal components. Samples were regressed against mitochondrial transcript content (as mitochondrial transcripts become enriched in unhealthy cells). There was no significant hemoglobin signature suggestive of red blood cell contamination in either dataset.

Once integrated, cell groups were identified by co-expression of canonical markers. Briefly, we identified clusters containing pulmonary artery endothelial cells (PAEC, *EFNB2, HEY1+*), pulmonary capillary endothelial cells (PCEC, *SPARC, SGK1+*), pulmonary venous endothelial cells (PVEC, *ACKR1, VWF+*), and lymphatic endothelial cells (LEC, *LYVE1+*) by comparing our gene expression data to a published lung endothelial cell dataset [30]. Due to the sparsity of single-cell sequencing data, clusters that were associated with specific celltype features were assumed to contain those cells. We were similarly able to identify epithelial subsets including AECI (*AGER*, *PDPN*, *HOPX+*), type-2 alveolar epithelial cells (AECII, *SFTPC, ABCA3, SLC34A2+*), basal cells (*TP63*, *SOX2, KRT5+*), goblet cells (*PIGR*, *SCGB1A1*, *SPDEF+*), ciliated cells (*PIFO*, *RFX3*, *DNAH2+*), and club cells (*FOXA1*, *SCGB1A1*, *SCGB3A2*+). Stromal cells such as fibroblasts (*VCAN+*), myofibroblasts (*ACTA2+*), alveolar macrophages (*ZEB2*, *CD68+*), non-resident macrophages (*MARCO*, *CD68*+), B cells (*MS4A1+*), T cells (*CD3E+*), NK/NKT cells (*NKG7, CD56+*), and mast cells (*TPSAB1*, *MS4A2+*) were similarly identified. Representative clusters are shown in Figure 1.

Differential expression analysis was performed by specifying celltype and contrasting by disease (“IPF” or “Control”), or, specifically using the University of Colorado dataset, by genotype at rs35705950 (variant “TT” vs. non-variant “GG”). All differential expression analysis was performed using the Seurat function “FindMarkers,” which by default uses a Wilcoxon rank sum test performed on corrected Poisson residuals derived from the SCTransform v2 pipeline. Reported differences are expressed as log_2_ fold-change of “reference” vs. “condition of interest” unless otherwise indicated. Reported significance values (*q*) are adjusted for a 5% false-discovery rate (FDR).

Single-cell RNA sequencing populations from the larger Vanderbilt dataset were mapped to the University of Colorado dataset. Celltype predictions were made using Seurat [28] on the basis of the reference Vanderbilt dataset using the “FindTransferAnchors” function followed by the “TransferData” function, which applied predictions to the query University of Colorado dataset. Predicted celltypes in the University of Colorado dataset were used for downstream analysis. To harmonize visualizations, the top 30 reference principal components were projected from the Vanderbilt dataset onto the University of Colorado dataset using the MapQuery function in Seurat.

### 2.3. Cell Quantification

After assigning celltype identities to each cell or nucleus in each sample as above, numbers of cells per celltype per sample were normalized to the average number of cells per celltype per sample. Samples were then partitioned by disease (Vanderbilt and Colorado datasets), or by genotype (Colorado dataset). For the Vanderbilt dataset, statistical comparisons were made using Students’ *t*-test with Welch’s correction, which were then converted to FDR *q* values. For the Colorado dataset, the small sample size of the dataset precluded statistical analysis of proportions for each celltype (only 2 donors per genotype per group).

### 2.4. Cell–Cell Communication

To facilitate cell signaling analysis, we used the NicheNet R package (‘nichenetr’, version 1.1.0, https://github.com/saeyslab/nichenetr, Ghent, Belgium, accessed on 28 March 2022) [27], which uses a curated, incident-ligand-based, transcriptional signature approach to identify ligand activity in receiving cells, then filters ligand candidates by receptor presence on the receiver and ligand expression from sending cells. Moreover, NicheNet enables differential analysis of gene expression and is flexible to the variable abundance of celltypes in single-cell and single nuclear sequencing. For the Vanderbilt dataset, we followed the standard NicheNet analysis pipeline (https://github.com/saeyslab/nichenetr/blob/master/vignettes/seurat_steps.md, accessed on 30 April 2022) stratifying samples according to diagnosis (“IPF” or “Control”). For the Colorado dataset, we stratified samples according to diagnosis and genotype (“GG” or “TT”). NicheNet uses a gene enrichment process requiring estimation of the expected gene counts on the basis of features present within the dataset. Generally, features within the “ligand_target_matrix” are filtered to those genes within the recovered transcriptome and set as background; however, due to the sparseness of single nuclear datasets, which biases the most highly transcriptionally active genes, the Colorado cohort was analyzed using all genes within the “ligand_target_matrix” as background. For the purposes of this study, the receiver celltypes of greatest interest were AECIs. Sender celltypes included AECI—to evaluate for autocrine signaling—as well as AECII, basal cells, ciliated cells, club cells, goblet cells, endothelial cells, fibroblasts, macrophages, lymphocytes, and Mast cells, unless there were insufficient cell numbers (minimum set to 3) in the reference or affected condition, in which case those celltypes were omitted.

### 2.5. Immunohistochemistry

Formalin fixed, paraffin-embedded, deidentified lung tissue specimens (*n* = 16 control, *n* = 47 IPF) were collected from the NHLBI-sponsored Lung Tissue Research Consortium or from the University of Colorado Hospital (COMIRB #15-1147). Control specimens were generally obtained from lungs rejected for transplant. IPF diagnosis was made by previously published American Thoracic Society criteria [31] and was independently adjudicated by a multidisciplinary panel at the University of Colorado on the basis of available clinical, pathologic, and radiologic data. Specimens were previously genotyped for the *MUC5B* variant (SNP rs35705950 G; T) using Taqman probes (ThermoFisher Scientific, Waltham, MA, USA, SNP microarray, or sequencing of donor DNA [7,8,9]. Control and IPF samples were similar in age (average age in controls 63.4, average age in IPF 64.1, *p* = 0.72) and smoking history (11/14 controls, 38/57 IPF were former smokers, *p* = 0.52 by Fisher’s exact test). Samples obtained from donors who were current smokers, had autoimmune lung disease, or had COPD were excluded.

Tissues were deparaffinized in xylene and ethanol and antigen retrieval was undertaken by heating for 27 min in sodium citrate buffer (10 mM, pH 6.0). Once cooled, samples were blocked for 1 h at room temperature with phosphate-buffered saline containing 2.5% bovine serum albumin (ThermoFisher Scientific, Waltham, MA, USA). Samples were stained overnight at 4 °C with polyclonal rabbit anti-ADAM17 (Proteintech, Rosemont, IL, USA, cat. 20259-1-AP) or unimmunized rabbit IgG (Sigma-Aldrich, St. Louis, MO, USA), washed, and stained with Alexa Fluor 488 donkey anti-rabbit IgG (Jackson ImmunoResearch Laboratories, Inc, West Grove, PA, USA) and DAPI (Sigma-Aldrich, St. Louis, MO, USA). Random 20× images of each tissue were taken on a Keyence BZ-X800 fluorescent microscope (Keyence Corporation of America, Itasca, IL, USA) without regard to specific tissue features. Exposure times were set to minimize background staining on the basis of rabbit isotype controls. Monochrome images for each channel were exported and analyzed using the ImageJ software platform “Fiji” (https://github.com/fiji/fiji, Madison, WI, USA, accessed on 13 November 2019).

## 3. Results

### 3.1. IPF Dramatically Alters Celltypes in the Lung

To gain a better understanding of IPF pathobiology we analyzed publicly available, deidentified, single-cell and single nuclear datasets. We selected two datasets, one from Vanderbilt University (GSE135893) [25], composed of 10 control and 12 IPF donors, and another from our group at the University of Colorado (GSE161685), composed of 2 control and 2 IPF donors, stratified further by genotype for the *MUC5B* rs35705950 (G; T) variant [26]. We recalculated coefficients for L1-regularized, negative binomial regression, modeling each UMI count as a variable dependent on sequencing depth using SCTransform v2 [29,32]. After identifying cell clusters in the Vanderbilt dataset on the basis of canonical marker expression (Figure 1, see the Section 2), we were able to project celltype labels from the Vanderbilt dataset to the University of Colorado dataset, as well as to plot single nuclear sequencing clusters from the Colorado dataset onto the Vanderbilt UMAP scaffold by using regularized principal components (Figure 1).

We used celltype identities from the Vanderbilt dataset to predict cellular identities in the Colorado dataset. Median score for all predicted celltypes was 0.72 (IQR 0.56–0.85), with the highest scoring celltype being T cells (median 0.95, IQR 0.71–0.99) and the lowest scoring celltype being PAEC (median 0.42, IQR 0.34–0.48). AECI were in the upper end of the identity scoring range with median of 0.9 (IQR 0.73–0.94).

We examined cellular differences in IPF lung compared to the control (Figure 2), as well as in *MUC5B* variant (TT) and non-variant (GG) samples (Table 1). We noted significant differences in the cellular content of control and IPF lungs in both datasets. In the Vanderbilt dataset, we found a significant reduction in the number of AECI cells, with commensurate increases in club cells and basal cells (FDR-corrected *q* values <0.05). We observed nominal increases in B cells, myofibroblasts, and goblet cells, but these were not significant when corrected for multiple testing. We similarly observed a nominal decrease in PAEC cell content. There was a trend toward increased numbers of ciliated cells in the IPF samples, but this did not reach nominal statistical significance (*p* = 0.08).

We observed similar nuclear identity differences when we examined predicted celltypes in the Colorado dataset partitioned by disease state or *MUC5B* genotype (Table 1). Consistent with the Vanderbilt dataset, there was a dropout of AECI, as well as loss of AECII and PCEC. We also observed increases in B cell nuclei; plasma cell nuclei; and ciliated, goblet, and basal cell nuclei in IPF. When stratified by genotype, there were apparent increases in macrophage, plasma cell, AECII, goblet cell, club cell, and PCEC nuclei in the TT donors. Given the almost negligible numbers of AECI nuclei recovered from IPF lungs, we proceeded with cell signaling analysis in the Vanderbilt dataset.

### 3.2. AECI Cell Signaling Analysis in Control and IPF Lung: Single-Cell RNA-seq

To understand how the IPF environment influences AECI cell number and function, we analyzed the Vanderbilt dataset using the cell–cell signaling analysis package “nichenetr” [27]. NicheNet uses an inferential model based on prior understanding of how ligands influence gene expression in receiver cells, then selects from potential ligands on the basis of receptor expression by the receiver (in this case, AECI). NicheNet also examines ligand expression in sending cells within the local environment. Identification of true signaling pathways has been shown to be proportional to the Pearson correlation score [27].

We observed enrichment for a number of senescence, inflammation, and apoptosis-related genes in AECI cells from IPF cases in the Vanderbilt dataset (Figure 3a). Association analysis revealed ligand candidates among the top 20 Pearson scores, including SPP1, IL6 (and other IL6 family members, including OSM and LIF), IL1B, ADAM17, and EGFR family members such as TGFA (Figure 3a). Likely ligand–receptor pairs were determined by analysis of AECI receptor expression and curated interaction evidence [27] (Figure 3b). Analysis of the transcriptomes of other celltypes in the IPF lung revealed sources of incident signaling (Figure 3c). Likely source cells included macrophages, fibroblasts, myofibroblasts, endothelial cells, and alveolar epithelia; several of these celltypes were noted to be increased in IPF tissue.

Incident signals may be derived from a complex mixture of “source” cells. To confirm cell signals across conditions, we compared the inferred ligands to aggregated gene expression across all cell types present in IPF or control samples in the Vanderbilt dataset. Reasoning that relatively rare cell populations, such as fibroblasts, may make a negligible impact on gene expression aggregated over all cells derived from IPF or Control lungs, but may still have a large impact on neighboring cells, we ignored minimal fold-change limits in these differential expression results. We found general agreement between the inferential analysis in NicheNet and gene expression changes for ligands across all celltypes stratified by condition (Table 2). Notable exceptions included *ADAM17*, *VWF*, and *OSM*, which demonstrated a negative fold-change overall (Table 2). *ADAM17* appeared to be upregulated by fibroblasts and AECI but was downregulated by macrophages and alveolar macrophages in this dataset. *VWF* was downregulated by PAEC, PCEC, and PVEC, but upregulated by LEC. *OSM* was upregulated by alveolar macrophages but downregulated by other cells (Figure 3c). *AREG* was among the likely ligands but was not within the top 20 prioritized genes (average log_2_ fold-change in IPF vs. control 2.3968, *q =* 1.29 × 10^−14^). Moreover, expression of *CTGF* was also increased, but it was not represented within the top 20 prioritized genes (average log_2_ fold-change in IPF vs. control 0.4458, *q* = 3.15 × 10^−111^).

### 3.3. AECI Cell Signaling Analysis in Control and IPF Lung: Single Nuclear RNA-seq

To determine whether observed single-cell signaling pathways in the Vanderbilt dataset were common to other single-cell IPF datasets, and to extend our findings to the *MUC5B* variant genotype, we used NicheNet to analyze the Colorado dataset. Due to the sparseness of single nuclear data, we set the background of expressed genes to all genes contained within the NicheNet “ligand_target_matrix,” rather than filtering these only for genes that could be detected [27]. In general, we found broad agreement among inferred ligands between the Vanderbilt dataset and the Colorado dataset when we compared IPF to control tissues (Figure 4). We observed enrichment for signatures of SPP1, ADAM17, CTGF, FGF7, AREG, and TNFSF10, as well as others, but did not find support for *SPP1*, *ADAM17,* or *CTGF* differential expression on the basis of aggregate expression (Table 3). Review of cell-specific expression revealed upregulation of *SPP1* almost exclusively by non-alveolar macrophages, and these data were significant when this celltype was specified (average log_2_ fold-change 1.0290, *q* = 8.5428 × 10^−43^). *ADAM17* was upregulated by macrophages, AECI, and AECII, but downregulated by fibroblasts and myofibroblasts in IPF lung. Conversely, *CTGF* was generally downregulated or neutral by most examined cells, save PAEC, which upregulated it slightly.

We also noted the absence of inflammatory genes including *IL6*, *TNF*, and *IL1B* in the Colorado dataset. These genes were recovered at extremely low rates from the single nuclear dataset compared to single-cell data (*IL6* in Vanderbilt dataset, 3.2–4.9%, Colorado dataset 0.1%; *TNF* in Vanderbilt dataset 8.6–12.6%, Colorado dataset 1–3%; *IL1B* in Vanderbilt dataset 12.9–16.6%, Colorado dataset 0.1–0.2%) and were filtered out of the analysis. Nevertheless, given the otherwise consistent ligand enrichment between IPF cases in the Colorado dataset and IPF cases in the Vanderbilt dataset (Figure 4), we decided to evaluate expression of candidate ligands on the basis of *MUC5B* variant genotype.

### 3.4. AECI Cell Signaling Analysis in MUC5B Variant Carriers

We confirmed that *MUC5B* expression was increased across aggregated cells in variant vs. non-variant carriers (average log_2_ fold-change 0.1152, *q* = 4.8330 × 10^−18^); however, the adjusted difference specifically in AECI was not significant (average log_2_ fold-change 0.0287, *p* = 0.001, *q* = 1), suggesting AECI are not intrinsically influenced by the *MUC5B* variant. We also observed considerable attrition of AECI cells among IPF cases in the Colorado cohort (only 1% of the AECI counts in controls), and due to the small sample size, these comparisons were skewed toward the control donor in each group. Nevertheless, using aggregated comparisons (expressed genes across all “TT” vs. “GG” cells, independent of identity), we found that *ADAM17*, *CTGF*, and *AREG* were upregulated in *MUC5B* variant samples relative to nonvariant samples. The major contributor of *ADAM17* expression in *MUC5B* variants were myeloid-derived cells and AECII. AECI also upregulated *ADAM17*, suggesting autocrine signaling (average log_2_ fold-change 0.1379). Likewise, the major contributor of *CTGF* expression in variant-derived nuclei were fibroblasts and myofibroblasts. Finally, *AREG* was upregulated considerably by Mast cells in the TT lung, as well as AECII (average log_2_ fold-change 0.6948 and 0.3483, respectively).

### 3.5. Receptor Signatures among AECI in IPF and MUC5B Variant Carriers

We performed a secondary analysis to determine whether receptor upregulation could account for the NicheNet inferences of ligand activity in target AECI cells stratified by disease or genotype. Due to its potential pathobiologic relevance, we returned to examining the expression of *TNFSF10*, the TRAIL ligand, and its receptors in IPF in the Vanderbilt dataset. While Pearson inference suggested enrichment for TNFSF10 signaling in IPF AECI, the average log_2_ fold-change was negative and non-significant (Pearson 0.0693, average log_2_ fold-change −0.1873, *q* = 1). However, receptor expression for TNFSF10 is increased in IPF AECI (*TNFRSF10B*, average log_2_ fold-change 0.3659; *TNFRSF10C*, average log_2_ fold-change 0.4428) supporting a role for TNFSF10 signaling. Moreover, *TNFSF10* ligand expression was increased in a celltype-specific fashion in IPF by Mast cells (average log_2_ fold-change 0.7707), PCEC (average log_2_ fold-change 0.2835), AECII (average log_2_ fold-change 0.1631), ciliated cells (average log_2_ fold-change 0.0831), club cells (average log_2_ fold-change 0.4200), goblet cells (average log_2_ fold-change 0.1856), and basal cells (average log_2_ fold-change 1.0982). Conversely, we did not find evidence of significant variable expression of receptors for IFNG, OSM, or TGFA, although these were highlighted by NicheNet in initial analysis (Figure 3, Table 2).

To determine whether receptor rather than ligand upregulation could account for pathway activity in *MUC5B* variants, we analyzed receptor expression for AECI in the Colorado dataset. *SPP1*, *FGF7*, and *TNFSF10* were downregulated in *MUC5B* variant (TT) relative to non-variant (GG) nuclei. We found that, although *FGF7* expression was downregulated (average log_2_ fold-change −0.1196, *q* = 2.2011 × 10^−9^), particularly in fibroblasts (average log_2_ fold-change −0.9590), receptor expression on AECI was upregulated (*FGFR2* log_2_ fold-change 0.0573, *q* = 1; *DDR1*, log_2_ fold-change 0.1213, *q* = 0.0055). Moreover, we observed that, while *TNFSF10* expression, on average, decreased (average log_2_ fold-change −0.0608, *q* = 2.82 × 10^−06^), similar to the Vanderbilt dataset, expression by PCEC and AECII was increased (average log_2_ fold-change 0.3439, and 0.1545, respectively). Receptor expression remained unchanged. These data support a possible role for FGF7 and TNFSF10 signals in *MUC5B* variant-derived AECI.

### 3.6. ADAM17 Is Enriched in MUC5B Variant Carriers

The matrix metalloproteinase ADAM17 acts upstream of AREG, LIF, TGFα, TNFα, OSM, and IL6 trans-signaling [33]. Given the presence of these signals in IPF lung (Figure 3 and Figure 4), the importance of AREG and IL6 pathways in IPF [22,23,24,34,35,36,37,38], and the finding that *ADAM17* signals were increased in AECI from *MUC5B* variants (Figure 4), we hypothesized that excessive *ADAM17* signals in *MUC5B* variants could precipitate development of IPF. We found that ADAM17 staining intensity was increased among *MUC5B* variant carriers in lung tissues derived from non-IPF donors (Figure 5). These data confirm that ADAM17 is increased in prefibrotic *MUC5B* variant lungs.

## 4. Discussion

In this report, we utilized publicly available single-cell and single nuclear datasets to investigate cell–cell communication in the IPF lung. We found, consistent with our previous data [22,24], that the EGFR ligand AREG and the cytokine IL6 may play significant complimentary roles in influencing AECI loss in IPF. We also find evidence of roles for TNF, IL1B, LIF, SPP1, FGF7, and TNFSF10 on AECI biology (Figure 6). Moreover, we found that, among single nuclei isolated from *MUC5B* variant vs. non-variant lungs, signatures of AREG, ADAM17, and CTGF were increased. Given our recent data [22] and [24], the finding that *AREG* expression and signaling in AECI associates with *MUC5B* genotype suggests a pathophysiologic association contributing to IPF development. We now extend these findings by showing that the *MUC5B* promoter variant is associated with increased *ADAM17* expression and signaling in non-fibrotic lungs.

The emergence of ADAM17 as a significant contributor to AECI cell signals on the basis of the *MUC5B* variant could have implications for disease pathobiology. ADAM17 is involved in a variety of inflammatory processes, including activation and release of TNF, modulation of neuregulins such as AREG [39], and being a necessary cofactor for IL6 trans-signaling [33,40], which has been shown to promote pulmonary fibrosis in bleomycin injury [23,41,42]. Importantly, *ADAM17* has been shown to be upregulated by endoplasmic reticulum stress [41], which is a proposed mechanism of *MUC5B* variant-dependent IPF [42,43].

Recently, some researchers have described the presence of “aberrant basaloid” cells in IPF [25,44]. Other researchers have shown that incident cytokine and morphogen signals acting on AECII may promote transdifferentiation of AECII to basaloid and airway lineages, which overlap with *KRT5* [45] and *KRT17* expression [46,47]. AECI examined in this dataset do not express *KRT5* or *KRT17* (see Appendix A); however, we can corroborate increases in the prevalence of basaloid cells in IPF cases among the Vanderbilt and Colorado datasets and have likewise identified signatures of airway metaplasia (Figure 2, Table 1). Moreover, researchers have described a pre-alveolar transitional cell state (PATS) characterized by *KRT19* expression [48]. The AECI in this manuscript largely expresses *AGER*, a marker of AECI, and as a consequence do not express *KRT19* (see Appendix A). Nevertheless, we find that AECI in IPF do express *KRT8* (see Appendix A). *KRT8* has been flagged by researchers as a marker of impaired AECI differentiation [49,50] and is shared by the transient AECI differentiation states described above (basaloid and PATS). Our work does not specifically identify which cytokines or morphogens may be involved in differentiation of these transient subsets, but we note that the attrition of AECI appears to correlate with IL6 and EGFR signaling pathways. Further investigation into whether IL6 or AREG signals are sufficient or for AECI loss, and whether aberrant or transdifferentiation of AECII is responsible for AECI cell loss, will be necessary.

To our knowledge, this is among the first applications of NicheNet to single-cell signaling in the lung. NicheNet analysis combines the strengths of gene coexpression approaches to define signaling ligands with single-cell approaches to specifically define receptor and ligand pairs. The results of this analysis were broadly consistent across two very different datasets, highlighting the robustness of the approach. Moreover, our analysis specifically targeted a celltype—AECI—which demonstrates marked attrition from diseased tissues. Due to their highly interconnected anatomy, AECI are difficult to disaggregate from the lung, advantaging single nuclear RNA sequencing approaches for reliable study. Other cell types within the lung are also amenable to NicheNet analysis, including ciliated cells, which were not numerically significantly increased in IPF in this investigation but have been found to be increased in IPF lung elsewhere [51]. Additional targets amenable to the NicheNet approach in future studies include PCEC, which, commensurate with loss of AECI, were markedly missing from IPF tissues.

Limitations of this study include the small numbers of AECI recovered from IPF tissue and discrepancies between single nuclear RNA sequencing and single-cell RNA sequencing, which limited our ability to confirm *IL1B*, *TNF*, and *IL6* differential expression in IPF in the Colorado dataset. *TNF* mRNA in particular is well known to be translationally regulated; whether mRNA stability drives the discrepancy in transcript abundance between cells and nuclei in the two datasets is unclear. Other authors have shown that inflammatory genes are relatively enriched by single-cell preparation vs. nuclear RNA-seq [52]; however, they have attributed this enrichment of inflammatory transcripts among single cell RNA-seq studies to proteolytic processing necessary for cell isolation, rather than a byproduct of subcellular localization. Other inconsistencies across the datasets may be related to heterogeneity of disease in available tissue, lack of AECI in IPF samples to make these studies more robust, and the significant size limitations of the Colorado dataset. Finally, the data described here are largely inferential; further experiments to confirm presence or absence of ligands apart from ADAM17, presence of ADAM17 enzymatic activity among variants, and gain-of-function/loss-of-function studies will be necessary to confirm these results.

## 5. Conclusions

Single-cell and single nuclear RNA sequencing validate roles for AECI attrition contributing to pulmonary fibrosis. Signatures of AREG, IL6, and TNFSF10 signaling in fibrotic lung suggest possible mechanisms of disease maintenance and AECI cell loss. Moreover, in *MUC5B* variant carriers, expression of *AREG* and *ADAM17* could contribute to the pathophysiology of lung injury. The role of these pathways and their relevance for AECI attrition will require further development of appropriate gain-of-function and loss-of-function models of AECI biology.

## Figures and Tables

**Figure 1 cells-11-03319-f001:**
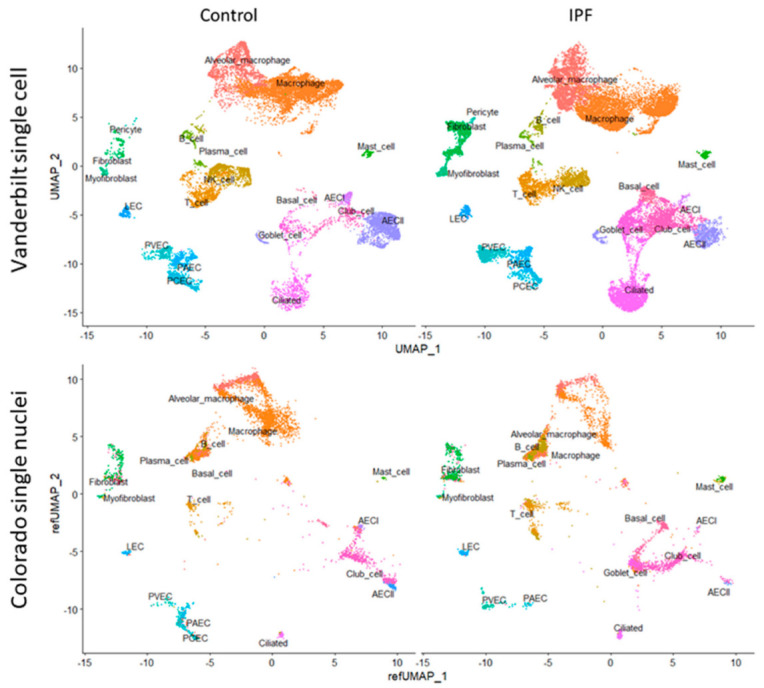
(**Top**) UMAP clustering of the Vanderbilt IPF single-cell RNA sequencing dataset (top) stratified by diagnosis (“Control” vs. “IPF”). (**Bottom**) Colorado single nuclear RNA sequencing dataset clustered by projecting principal components onto the Vanderbilt UMAP (“refUMAP”) and transferring celltype labels.

**Figure 2 cells-11-03319-f002:**
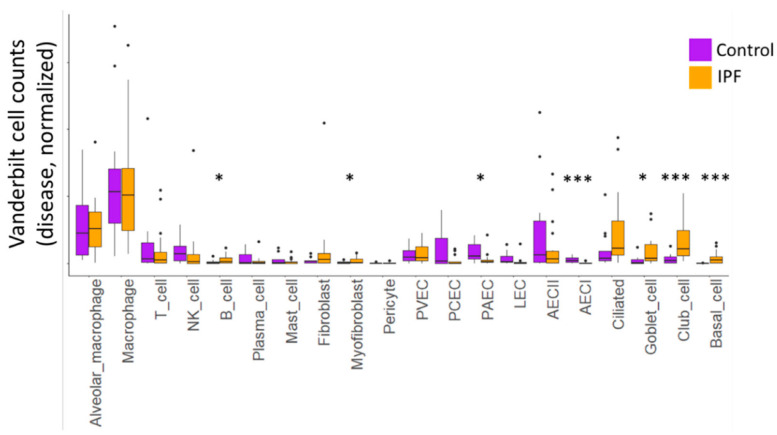
Box-and-whiskers plot of normalized cell counts from the Vanderbilt dataset. Black points represent samples whose celltype proportions fall outside of the interquartile range. Asterisks represent nominal *p*-values within each cell type comparing 10 control to 12 IPF samples using Students’ *t*-test with Welch’s correction. *p* < 0.05 (*), and *p* < 0.005 (***). FDR-corrected values for (***), *q* < 0.05.

**Figure 3 cells-11-03319-f003:**
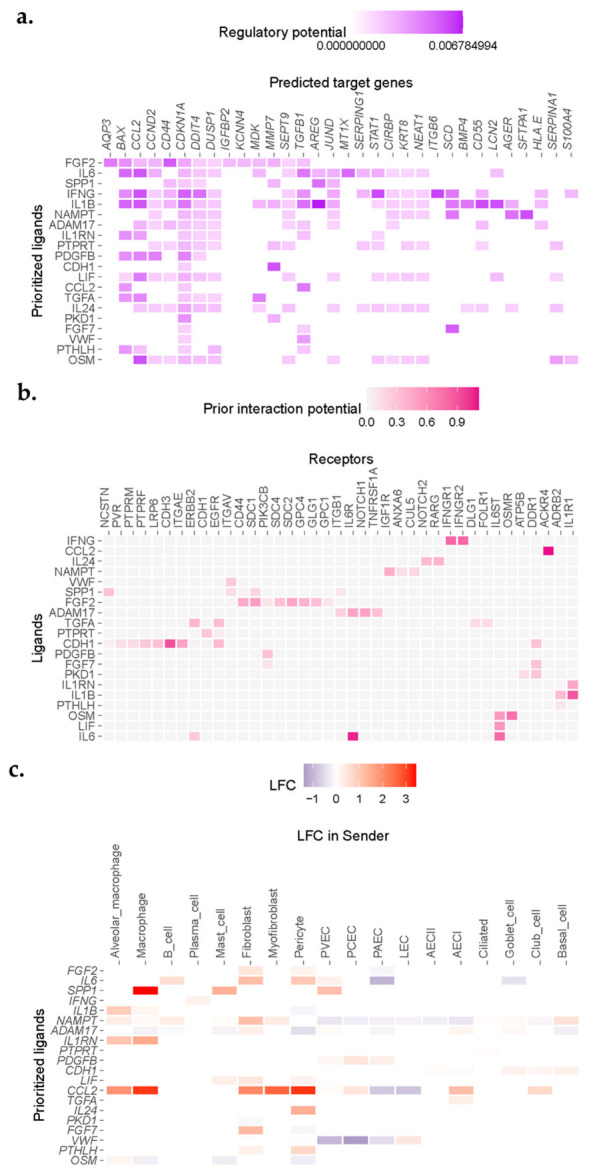
NicheNet analysis revealed enrichment for IL1, IL6, and EGFR ligand signaling in single cells sequenced from IPF donors. (**a**) Heatmap demonstrating correlation scores for regulation of genes (columns) by potential ligand (rows). (**b**) Ligand−receptor heatmap demonstrating strength of predicted interaction of ligands and receptors filtered by expression from (**a**). (**c**) Expression of putative ligands by celltypes neighboring AECI in fibrotic lung.

**Figure 4 cells-11-03319-f004:**
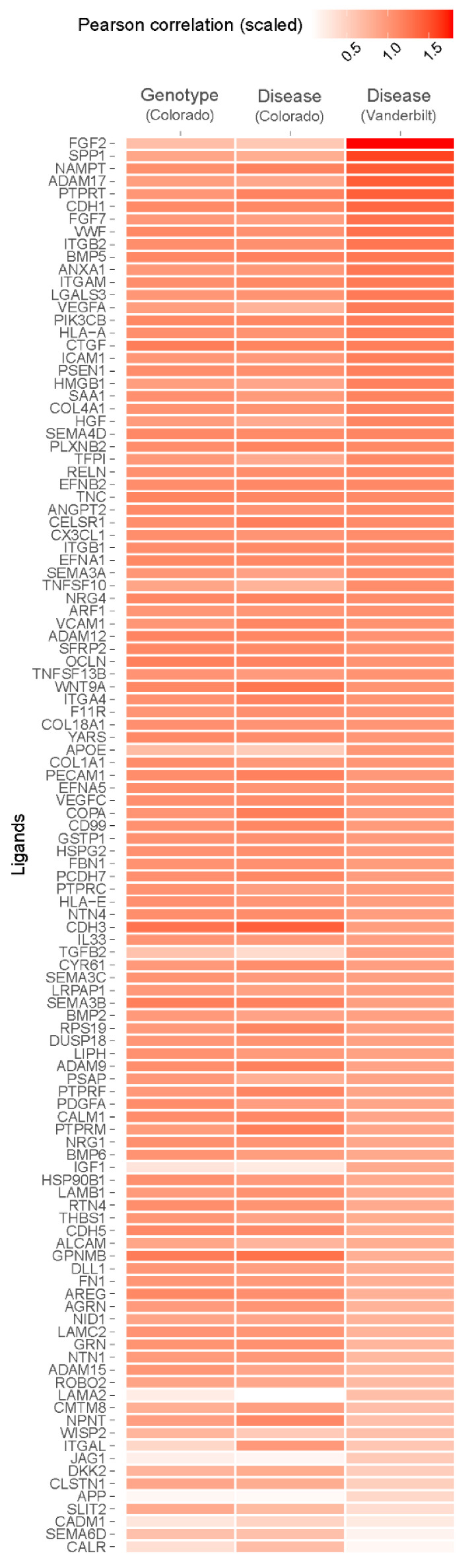
Heatmap depicting Pearson correlation enrichment scores based on NicheNet analysis of AECI from *MUC5B* variant (TT vs. GG, “Genotype”) and IPF vs. control (“Disease”) samples from the Colorado single nuclear RNA sequencing dataset, and IPF vs. control samples (“Disease”) from the Vanderbilt single-cell RNA sequencing dataset. Samples are scaled and sorted according to correlation score from the Vanderbilt dataset.

**Figure 5 cells-11-03319-f005:**
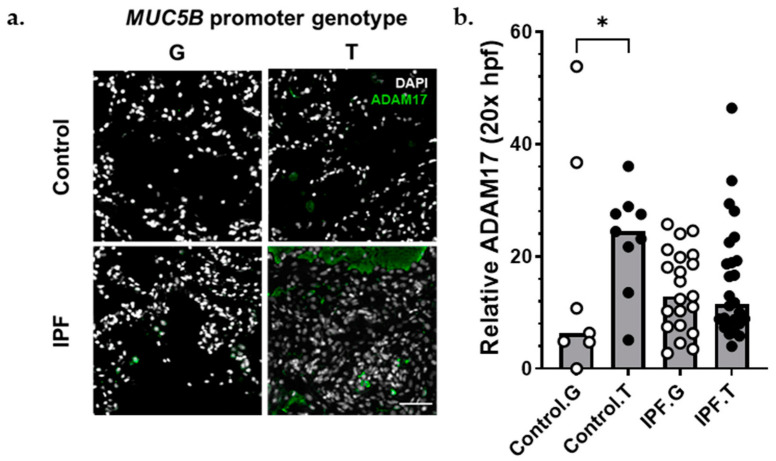
ADAM17 protein was increased among *MUC5B* variant carriers. (**a**) Representative immunofluorescence images of ADAM17 expression (green) among control (**top**) and IPF (**bottom**), *MUC5B* nonvariant (G), and *MUC5B* variant (T) carriers. (**b**) Semiquantitative analysis of ADAM17 staining normalized to number of nuclei per 20× field. Each bar represents the median of acquired data with *p* < 0.05 (*, Kruskal–Wallace test). Scale bar (**bottom**, **right**) = 50 μm.

**Figure 6 cells-11-03319-f006:**
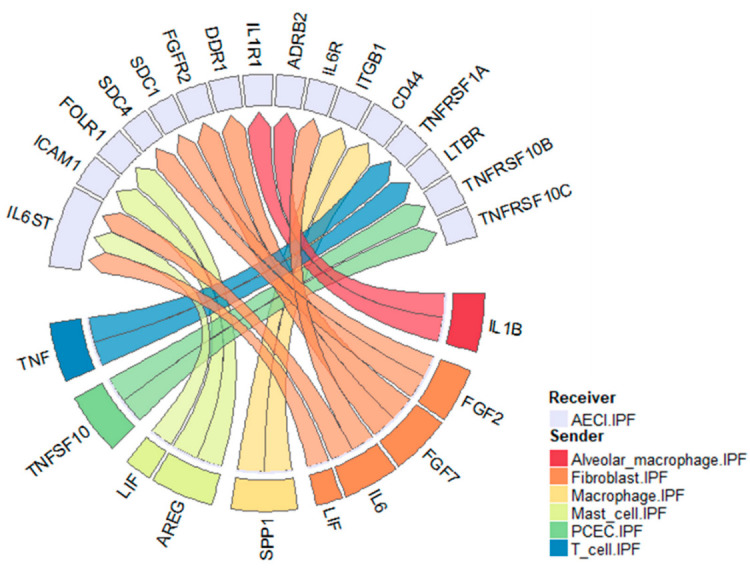
Circos plot of ligand–receptor pairs impacting AECI in the IPF lung. Data derived from the Vanderbilt single-cell dataset. Selected ligands represent those enriched in IPF over control. Receptors represent those expressed by AECI. Colors represent sending celltypes as indicated by the legend.

**Table 1 cells-11-03319-t001:** Predicted celltype numbers according to single nuclear RNA sequencing partitioned by disease and *MUC5B* genotype in the Colorado dataset.

Predicted Celltype	Control	IPF	GG	TT
Alveolar_macrophage	1013	625	950	688
Macrophage	3223	1505	1441	3287
T_cell	158	424	197	385
NK_cell	0	0	0	0
B_cell	637	2793	1515	1915
Plasma_cell	51	616	89	578
Mast_cell	16	106	76	46
Fibroblast	207	466	320	353
Myofibroblast	27	45	20	52
Pericyte	0	0	0	0
PVEC	28	100	54	74
PCEC	350	0	37	313
PAEC	67	17	27	57
LEC	34	96	89	41
AECII	569	10	4	575
AECI	859	9	520	348
Ciliated	152	924	650	426
Goblet_cell	0	233	50	183
Club_cell	1145	873	438	1580
Basal_cell	346	703	566	483

**Table 2 cells-11-03319-t002:** Relationships between Pearson correlation coefficients and log_2_ fold change (entire Vanderbilt dataset) for predicted active ligands impacting AECI cells. Ligands ordered according to Figure 3b.

Ligand	Pearson	mRNA Fold-Change (log_2_, All Cells)	*q* (FDR)
FGF2	0.1157	0.0367	NA
CDH1	0.0877	0.1666	1.02 × 10^−44^
IFNG	0.0967	0.2642	0.5278
TGFA	0.0856	0.0454	NA
SPP1	0.1025	2.4719	0
ADAM17	0.0925	−0.1084	1.02 × 10^−10^
VWF	0.0836	−0.4857	1.91 × 10^−08^
IL1RN	0.0914	0.8682	1.28 × 10^−184^
IL1B	0.0934	0.3688	1.52 × 10^−18^
LIF	0.0861	0.0596	2.44 × 10^−21^
IL6	0.1039	0.1094	1.07 × 10^−11^
OSM	0.0833	−0.0647	1

**Table 3 cells-11-03319-t003:** Differential expression results for selected inferred ligands in the Colorado single nuclear dataset. Predicted ligands ordered based on Pearson score in the Vanderbilt dataset. Fold-change (log_2_) represents all IPF samples compared to control.

Ligand	Pearson	Fold-Change (log_2_, All Cells)	*q* (FDR)
SPP1	0.0238	0.1883	1
ADAM17	0.0254	−0.2476	3.78 × 10^−61^
AREG	0.0314	0.0776	9.23 × 10^−5^
CTGF	0.0338	−0.0559	1.45 × 10^−25^
FGF7	0.0277	0.0838	0.0831
TNFSF10	0.0225	0.0715	9.71 × 10^−20^

## Data Availability

All data used for this analysis are publicly available through the National Library of Medicine Gene Expression Omnibus repository (https://www.ncbi.nlm.nih.gov/gds, accessed on 19 July 2020). The Vanderbilt dataset was previously published and can be found under accession number GSE135893 (accessed on 9 December 2019). The Colorado dataset was previously published and can be found under GSE161685 (accessed on 17 December 2020).

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
