# Peer review of "Dysregulated Cell–Cell Communication Characterizes Pulmonary Fibrosis"

_cells, 2022, doi:10.3390/cells11203319_

Round 1

Reviewer 1 Report

The paper by Kurche et al. uses publically available data to demonstrate how AECI cells are controlled by the ligands produced by other cells types in IPF and MUC5B variant lungs. Using AECI transcriptomics, the authors found enrichment of IL6 and AREG signature in aforementioned cells stressing the pivotal role of IL-6 and AREG signalling in aberrant behaviour of epithelial cells in fibrotic lungs. The study is well conducted and addresses clinically relevant aspects including identification of alterations in cellular composition in fibrotic versus control lungs and cell-cell communication networks active during fibrosis progression. I have only minor comments to the authors .

1.       Please provide detailed description of patient cohorts. In addition, please provide information which regions of the lung were used for transcriptome analysis (as this could have major impact on cellular heterogeneity/composition detected).

2.       There are some differences in cellular composition of IPF and TT variant lungs. For example the numbers of club and basal cell go down in IPF lungs, however, they go up in the lungs of TT variant. How do the authors explain this?

3.       Several intermediate cell types have been describe in the lungs of IPF patients. How do you group/subgroup these cells in the analysis performed?

4.       Detection of the receptor on the particular cell type and the overall ligand abundance in IPF lungs does not always associate with the activity of the rescpective molecular pathway. Thus, these results have to be interpreted with caution and supported by the detailed characterisation of downstream signalling mediators.

Author Response

Reviewer 1:

  1. Please provide detailed description of patient cohorts. In addition, please provide information which regions of the lung were used for transcriptome analysis (as this could have major impact on cellular heterogeneity/composition detected).
  • The Vanderbilt samples are composed of deceased donor lung specimens rejected for transplant or IPF lungs explanted at time of transplant at the Norton Lung Institute in Phoenix AZ, or Vanderbilt University. For control lungs, multiple regions within 2cm of the pleura were harvested for single cell digestion. For IPF lungs, multiple regions representing areas of disease involvement were harvested.
  • Demographics for the Vanderbilt dataset are as follows:

Control (10)

IPF (12)

P value

Age (range)

35.2 (17-54)

65.17 (56-74)

1.135x10-5 (Students' t)

Sex (M/F/Unknown)

7/2/1

7/5/0

0.6424 (Fisher)

Race (white/nonwhite)

6/1/3

9/3/0

0.9999

Tobacco (Y/N/Unknown)

7/1/2

6/6/0

0.1577

  • The Colorado dataset samples are composed of whole lung explants or peripheral biopsy specimens from the University of Pittsburgh or the Lung Transplant Research Consortium, flash frozen from buffered medium and stored at -80ÌŠC from the time of harvest. Participants or kin were consented for genetic studies at the time of harvest. Frozen tissue fragments were taken from peripheral tissue when identifiable.
  • Demographics for the Colorado dataset are as follows:

ID

1097743

1097858

1058227

1097691

Diagnosis

normal

normal

ipf

ipf

Genotype

GG

TT

GG

TT

Age

64

64

65

64

Sex

male

male

male

male

Race

white

white

white

white

Tobacco

Unknown

Unknown

Unknown

Unknown

  1. There are some differences in cellular composition of IPF and TT variant lungs. For example the numbers of club and basal cell go down in IPF lungs, however, they go up in the lungs of TT variant. How do the authors explain this?
  • In the Vanderbilt dataset, which contains multiple samples, there is a trend toward increased basal cells and increased club cells in IPF. The Colorado dataset demonstrates this effect for Basal cells, but not for Club cells. Conversely, when stratified by genotype, the proportion of Basal cells is increased in MUC5B variant derived samples, but not Club cells. The reviewer highlights this discordance.
  • This could be an artifact of sampling; due to the very limited sample size described in the Colorado dataset, compared to the Vanderbilt dataset, further studies are necessary to accurately determine the relationship between MUC5B genotype and cellular composition.
  • Alternatively, mistyping of inferred identities in the Colorado single nuclear dataset could give rise to this discrepancy. Accordingly, we find that median celltype prediction scores were lower for Basal cells (0.436) and Club cells (0.616) compared to other epithelia, including AECI (0.896) and Ciliated cells (0.895).
  • Of note, effects on AECI cells are more consistent across conditions, supporting our focus on these cells.
  • Finally, the differences could reflect intrinsic differences in cell recovery based on the two methods of isolation: Lung tissue digests in the Vanderbilt dataset, vs tissue disruption in liquid nitrogen and nuclear extraction in the Colorado dataset.

  1. Several intermediate cell types have been describe in the lungs of IPF patients. How do you group/subgroup these cells in the analysis performed?
  • The reviewer may be referring to the “aberrant basaloid” cell type described in Kropski et al., Kaminski et al., and confirmed by Tata and colleagues [1-3]. These cells have been observed to be defined in part by expression of the genes KRT5-/KRT17+. We do not see significant overlap with these cells among AGER+ AECI cells. [See attached file]
  • Alternatively, the reviewer may be referring to the observations by Peng et al. and Tata et al. referring to terminal airway progenitor cells [4,5]. Peng et al. describe behaviors of respiratory bronchial epithelial cells which are absent in mouse, have the plasticity to differentiate into AEC2, but express secretory markers such as SCGB3A2 [4]. Tata et al. describe a terminal airway progenitor cell which they term “AT0,” characterized by expression of both secretory (SCGB3A2) and surfactant proteins (SFTPB+, SFTPC+/-) and can bipotently generate either alveolar or secretory lineages. These cells are largely similar. We find that selection of cells for AGER+ expression, a known AECI marker, largely excludes these “AT0” cells from analysis (AECI, circle; SFTPB+ SCGB3A2+, arrowheads for overlap): [See attached file]
  • Finally, the reviewer may be referring to the KRT8+ intermediate cells which lie in the transition between ATII and ATI [6,7]. We find evidence for coexpression of KRT8 within the examined cohort, however based on observation it appears as though these coexpressing cells may be disproportionately lost in IPF. We would expect KRT8 expression to be synonymous with these cells under pathologic conditions (circles). Future investigation is indicated. [See attached file]
  1. Detection of the receptor on the particular cell type and the overall ligand abundance in IPF lungs does not always associate with the activity of the rescpective molecular pathway. Thus, these results have to be interpreted with caution and supported by the detailed characterisation of downstream signalling mediators.

We agree and have included changes in the discussion to reflect this.

References

  1. Habermann, A.C.; Gutierrez, A.J.; Bui, L.T.; Yahn, S.L.; Winters, N.I.; Calvi, C.L.; Peter, L.; Chung, M.I.; Taylor, C.J.; Jetter, C.; et al. Single-cell RNA sequencing reveals profibrotic roles of distinct epithelial and mesenchymal lineages in pulmonary fibrosis. Sci Adv 2020, 6, eaba1972, doi:10.1126/sciadv.aba1972.
  2. Adams, T.S.; Schupp, J.C.; Poli, S.; Ayaub, E.A.; Neumark, N.; Ahangari, F.; Chu, S.G.; Raby, B.A.; DeIuliis, G.; Januszyk, M.; et al. Single-cell RNA-seq reveals ectopic and aberrant lung-resident cell populations in idiopathic pulmonary fibrosis. Sci Adv 2020, 6, eaba1983, doi:10.1126/sciadv.aba1983.
  3. Kobayashi, Y.; Tata, A.; Konkimalla, A.; Katsura, H.; Lee, R.F.; Ou, J.; Banovich, N.E.; Kropski, J.A.; Tata, P.R. Persistence of a regeneration-associated, transitional alveolar epithelial cell state in pulmonary fibrosis. Nat Cell Biol 2020, 22, 934-946, doi:10.1038/s41556-020-0542-8.
  4. Basil, M.C.; Cardenas-Diaz, F.L.; Kathiriya, J.J.; Morley, M.P.; Carl, J.; Brumwell, A.N.; Katzen, J.; Slovik, K.J.; Babu, A.; Zhou, S.; et al. Human distal airways contain a multipotent secretory cell that can regenerate alveoli. Nature 2022, 604, 120-126, doi:10.1038/s41586-022-04552-0.
  5. Kadur Lakshminarasimha Murthy, P.; Sontake, V.; Tata, A.; Kobayashi, Y.; Macadlo, L.; Okuda, K.; Conchola, A.S.; Nakano, S.; Gregory, S.; Miller, L.A.; et al. Human distal lung maps and lineage hierarchies reveal a bipotent progenitor. Nature 2022, 604, 111-119, doi:10.1038/s41586-022-04541-3.
  6. Jiang, P.; Gil de Rubio, R.; Hrycaj, S.M.; Gurczynski, S.J.; Riemondy, K.A.; Moore, B.B.; Omary, M.B.; Ridge, K.M.; Zemans, R.L. Ineffectual Type 2-to-Type 1 Alveolar Epithelial Cell Differentiation in Idiopathic Pulmonary Fibrosis: Persistence of the KRT8(hi) Transitional State. Am J Respir Crit Care Med 2020, 201, 1443-1447, doi:10.1164/rccm.201909-1726LE.
  7. Strunz, M.; Simon, L.M.; Ansari, M.; Kathiriya, J.J.; Angelidis, I.; Mayr, C.H.; Tsidiridis, G.; Lange, M.; Mattner, L.F.; Yee, M.; et al. Alveolar regeneration through a Krt8+ transitional stem cell state that persists in human lung fibrosis. Nat Commun 2020, 11, 3559, doi:10.1038/s41467-020-17358-3.

Reviewer 2 Report

In the manuscript, the authors found loss of type-I alveolar epithelia (AECI) characterizes the single cell RNA transcriptome in fibrotic lung, and validated the pattern of AECI loss using single nuclear RNA sequencing. Therefore, the study is novelty. However, all datas were from public database which definitively required experimental verification.

1. Please use flow cytometry technology to verify the distribution of immune cells in Figure 2 on the IPF mouse model.

2. MUC5B is a very important factor, and single-cell results should be further verified in MUC5B knockout mice.

3. The full text describes the relationship between some cell regulation and interaction in IPF disease. It is a descriptive article, and its clinical significance is unclear. Whether a group of relatively important cells can be identified, and drug intervention can be carried out to improve its clinical value.

4. Does MUC5B function in immune cell populations?

5. For patients with MUC5B mutation, the role of the two genes AREG and ADAM17 in the process of lung injury requires further discussion and experimental verification.

Author Response

Reviewer 2:

  1. Please use flow cytometry technology to verify the distribution of immune cells in Figure 2 on the IPF mouse model.

We have previously shown that the SFTPC-Muc5bTg mouse has exaggerated lung injury in multiple models [1,2]. The content and distribution of immune cells in Figure 2 shows only a relative increase in total B cell numbers in IPF samples relative to controls, all other immune populations identified are statistically similar. The bleomycin-injured mouse is not specifically pertinent to the scope of this work, which focuses on AECI in human IPF. Nevertheless, phenotyping immune populations in SFTPC-Muc5bTg and Muc5b-/- animals (and MUC5B variant and nonvariant persons) is of considerable interest and will be an avenue for future investigations.

  1. MUC5B is a very important factor, and single-cell results should be further verified in MUC5B knockout mice.

The reviewer suggests that we should attempt to confirm the absence of alveolar epithelial cell signaling pathways described here in bleomycin-injured Muc5b-/- animals, presumably most specifically ADAM17. We agree, however there are several specific issues which make Muc5b-/- mice an imperfect model of human disease. First, murine models require genotoxic insult, such as bleomycin injury, to develop fibrosis. We have already shown that the Muc5b-/- mouse is resistant to fibrosis secondary to bleomycin injury [1], and a presence or lack of ADAM17, AREG, or IL6 may be difficult to deconvolute from the effects of bleomycin more generally. Moreover, a number of authors have recently found that the differentiation of distal lung stem cells in the human is very different from that in the mouse [3,4], which may be pertinent for understanding the effects of the MUC5B promoter polymorphism on AECI biology.

  1. The full text describes the relationship between some cell regulation and interaction in IPF disease. It is a descriptive article, and its clinical significance is unclear. Whether a group of relatively important cells can be identified, and drug intervention can be carried out to improve its clinical value.

Agree. We have added language in the discussion to reflect this.

  1. Does MUC5B function in immune cell populations?

MUC5B is necessary for airway defense [5]. Some authors have proposed that differences in sialyation between MUC5B and other mucins may attract or deter airspace immune cells, in particular eosinophils [6]. Moreover, MUC5B expression is associated with airway defensin secretion, and the MUC5B gene polymorphism has been found to be associated with circulating neutrophil counts in PheWAS [7]. The mechanism of these effects are still unclear.

  1. For patients with MUC5B mutation, the role of the two genes AREG and ADAM17 in the process of lung injury requires further discussion and experimental verification.

This is correct vis a vis the gain-of-function MUC5B polymorphism. However, for further details on the relationships between AREG and IPF [8], and IL6, ADAM17, and IPF [9], please see the cited manuscripts.

References

  1. Hancock, L.A.; Hennessy, C.E.; Solomon, G.M.; Dobrinskikh, E.; Estrella, A.; Hara, N.; Hill, D.B.; Kissner, W.J.; Markovetz, M.R.; Grove Villalon, D.E.; et al. Muc5b overexpression causes mucociliary dysfunction and enhances lung fibrosis in mice. Nat Commun 2018, 9, 5363, doi:10.1038/s41467-018-07768-9.
  2. Kurche, J.S.; Dobrinskikh, E.; Hennessy, C.E.; Huber, J.; Estrella, A.; Hancock, L.A.; Schwarz, M.I.; Okamoto, T.; Cool, C.D.; Yang, I.V.; et al. Muc5b Enhances Murine Honeycomb-like Cyst Formation. Am J Respir Cell Mol Biol 2019, 61, 544-546, doi:10.1165/rcmb.2019-0138LE.
  3. Kadur Lakshminarasimha Murthy, P.; Sontake, V.; Tata, A.; Kobayashi, Y.; Macadlo, L.; Okuda, K.; Conchola, A.S.; Nakano, S.; Gregory, S.; Miller, L.A.; et al. Human distal lung maps and lineage hierarchies reveal a bipotent progenitor. Nature 2022, 604, 111-119, doi:10.1038/s41586-022-04541-3.
  4. Basil, M.C.; Cardenas-Diaz, F.L.; Kathiriya, J.J.; Morley, M.P.; Carl, J.; Brumwell, A.N.; Katzen, J.; Slovik, K.J.; Babu, A.; Zhou, S.; et al. Human distal airways contain a multipotent secretory cell that can regenerate alveoli. Nature 2022, 604, 120-126, doi:10.1038/s41586-022-04552-0.
  5. Roy, M.G.; Livraghi-Butrico, A.; Fletcher, A.A.; McElwee, M.M.; Evans, S.E.; Boerner, R.M.; Alexander, S.N.; Bellinghausen, L.K.; Song, A.S.; Petrova, Y.M.; et al. Muc5b is required for airway defence. Nature 2014, 505, 412-416, doi:10.1038/nature12807.
  6. Kiwamoto, T.; Katoh, T.; Evans, C.M.; Janssen, W.J.; Brummet, M.E.; Hudson, S.A.; Zhu, Z.; Tiemeyer, M.; Bochner, B.S. Endogenous airway mucins carry glycans that bind Siglec-F and induce eosinophil apoptosis. J Allergy Clin Immunol 2015, 135, 1329-1340.e1329, doi:10.1016/j.jaci.2014.10.027.
  7. Verma, A.; Minnier, J.; Huffman, J.E.; Wan, E.S.; Gao, L.; Joseph, J.; Ho, Y.-L.; Wu, W.-C.; Cho, K.; Gorman, B.R.; et al. <em>A MUC5B</em> gene polymorphism, rs35705950-T, confers protective effects in COVID-19 infection. medRxiv 2021, 2021.2009.2028.21263911, doi:10.1101/2021.09.28.21263911.
  8. Stancil, I.T.; Michalski, J.E.; Davis-Hall, D.; Chu, H.W.; Park, J.A.; Magin, C.M.; Yang, I.V.; Smith, B.J.; Dobrinskikh, E.; Schwartz, D.A. Pulmonary fibrosis distal airway epithelia are dynamically and structurally dysfunctional. Nat Commun 2021, 12, 4566, doi:10.1038/s41467-021-24853-8.
  9. Stancil, I.T.; Michalski, J.E.; Hennessy, C.E.; Hatakka, K.L.; Yang, I.V.; Kurche, J.S.; Rincon, M.; Schwartz, D.A. Interleukin-6–dependent epithelial fluidization initiates fibrotic lung remodeling. Science Translational Medicine 2022, 14, eabo5254, doi:10.1126/scitranslmed.abo5254.

Round 2

Reviewer 2 Report

Although there are database and literature data, the results of this manuscript still need to be verified by your own samples. Therefore, the experiments in questions 1, 2, and 5 need to be supplemented, otherwise the meaning found in data mining cannot be proved, so the meaning of this manuscript is limited.

Author Response

After discussion with Dr. Li and clarification from the Academic Editor, we have revised our manuscript to address concerns of the reviewer regarding the validation of differential ADAM17 expression in nondiseased, MUC5B variant lung tissue. We confirm that ADAM17, which is upstream of IL6, AREG, and TNF signaling, is differentially expressed in variant control tissues. This suggests that the MUC5B variant may predispose to IPF by promoting expression and/or activation of ADAM17, leading to downstream effects of TNF, IL6, and AREG. These data enhance the quality of this research considerably. 

We hope the reviewer will find these additions acceptable.